# The Impact of Social Jetlag on Sleep Quality among Nurses: A Cross-Sectional Survey

**DOI:** 10.3390/ijerph18010047

**Published:** 2020-12-23

**Authors:** Hyeonjin Kang, Miyoung Lee, Sun Joo Jang

**Affiliations:** 1College of Nursing, Eulji University, Daejeon 34824, Korea; kangmina104@naver.com (H.K.); mylee3730@eulji.ac.kr (M.L.); 2Red Cross College of Nursing, Chung-Ang University, Seoul 06974, Korea

**Keywords:** chronotype, sleep quality, shift work, social jetlag, nurses

## Abstract

Social jetlag (SJL) refers to an asynchrony between one’s chronotype and social working hours, which can be detrimental to health. The current SJL situation in shift nurses who work in dysregulation is poorly understood. Therefore, this study aimed to investigate SJL during night shifts and identify the predictors of sleep quality in rotating shift nurses. A cross-sectional study was conducted in 2018 on a sample of 132 shift-working nurses from two general hospitals in South Korea (the response rate was 88.8%). The SJL was measured with the Munich Chronotype Questionnaire, and sleep quality was measured with the Pittsburgh Sleep Quality Index and the Verran and Snyder–Halpern Sleep Scale. Data analysis was mainly based on a multiple regression, to identify SJL’s influence on nurses’ sleep quality. The average SJL during night shifts was 2 h and 3 min, and the average sleep duration during night shifts was 6 h 10 min. Multiple regression analysis revealed that SJL, day-shift fatigue, and sleep quality during night shifts affected nurses’ sleep quality. These variables accounted for 24.7% of the variance in overall sleep quality. The study concluded that overall sleep quality can increase with decreasing day-shift fatigue, decreasing SJL, and increasing sleep quality on night shifts.

## 1. Introduction

Shift work disturbs workers’ circadian rhythms. Therefore, it may reduce effective performance [1] and increase the incidence of occupational accidents and injuries, coronary heart disease, strokes, type 2 diabetes, cancer, obesity, depression, and all-cause mortality [2,3]. Furthermore, shift workers have been reported to be more depressed [4] and to show poorer cognitive functions than non-shift workers [5]. A prospective study of nurses reported that nurses who had worked shifts for more than 20 years had poorer cognitive functioning than nurses without shift work experience [6].

Shift work changes the awake and sleep time in a 24 h cycle [7] and it has an adverse impact on the amount and quality of sleep, as shift workers cannot avoid concerns about acute sleep loss during night and early morning shifts [2,8]. Adverse effects of poor sleep quality among nurses working shifts include daytime dysfunction, difficulty of recovery after sleep, poor sleep satisfaction, difficulty in sleep maintenance, difficulty with sleep initiation, and difficulty in waking up [9,10]. Previous studies have reported that nurses working shifts show signs of serious deterioration in sleep quality, as evidenced by the fact that 63% of nurses working shifts score beyond the cutoff for sleep disorders [11], and that approximately half to two-thirds of nurses working shifts experience poor sleep quality [12,13]. In particular, night-shift workers have to sleep during the day and, in such cases, sleep duration and quality are poorer compared to night sleep among day-shift workers. The consequent accumulation of sleep deprivation may hinder concentration and cause sleepiness during work hours [14].

Despite the continued reports of the adverse impact of shift work on nurses’ physical and mental health [8], shift work is inevitable for nurses because they must monitor patients, promptly respond to life-threatening situations, and continuously care for patients around the clock.

Amid such a reality, studies have examined individuals’ circadian rhythms and shift work in relation to chronotypes. Chronotypes are different types of circadian rhythms involved in sleep and wake cycles in humans, and individuals with different chronotypes show varying preferences for sleep–wake time and activity hours. Such individual differences manifest on a continuum from one’s circadian rhythm and are classified into two extremes: Morning and evening types. Studies have observed that evening chronotypes have more depressed and negative emotions with poor subjective well-being [15], while morning chronotypes generally have fewer frequent sleep disorders and higher tolerance to shift work [16,17,18]. Evening chronotypes experience increased sleep debt during the weekdays due to their work schedule beginning early in the morning, and they compensate for this by oversleeping on weekends (days off). Many people in our society exhibit changes in their sleep and wake cycles by up to a few hours between weekdays (work or school days) and weekends (days off).

Before the introduction of social jetlag (SJL) in the late 2000s [19], individuals were mostly classified based on chronotypes (morning-evening) in research. The construct of SJL refers to the asynchrony between one’s internal clock (circadian rhythm) and external clock (social work hours), that is, the misalignment of sleep–wake cycle and circadian rhythm. Studies have begun to examine the impact of SJL on sleep quality [20], the endocrine system, obesity [21], work ability, academic performance [22], physical health [23], mental health [24], and quality of life [25]. SJL can be better understood based on our understanding of jetlag experienced when traveling abroad.

According to previous studies on SJL, the view is that evening chronotypes report poor sleep quality and daytime fatigue, because their conventional social schedule is set around the time preferred by morning chronotypes and, thus, they accumulate substantial sleep debt throughout the week, depending on the severity of their SJL. This view is more common than those arguing that specific chronotypes experience fewer sleep disorders or have higher tolerance for shift work [26,27].

Therefore, it is necessary to study the impact of asynchrony between individuals’ circadian rhythm and chronotype, that is, SJL, on sleep quality among nurses for whom shift work is essential. In particular, it is important to understand the impact of SJL from night shifts, which completely counters the external environmental rhythm, thus causing sleep disorders, diminishing work performance, and reducing job satisfaction [28].

This study aimed to investigate the chronotypes and levels of SJL during night shifts among rotating shift nurses to analyze their sleep quality and identify the predictors of sleep quality. Based on these findings, we aimed to provide foundational data for developing efficient shift schedules by examining specific shift schedules that are easier to adapt to in relation to individual nurses’ chronotypes, SJL, and for interventions to promote effective work performance by improving sleep quality among nurses working shifts.

## 2. Materials and Methods

### 2.1. Study Design, Participants, and Ethical Considerations

This cross-sectional study aimed to examine the chronotypes among nurses working shifts and the level of SJL from night shifts to investigate the impact of SJL on sleep quality. Nurses working in two general hospitals in the Republic of Korea, who have worked 8-h rotating shifts, including at least two consecutive night shifts, for one year or longer were enrolled in this study. Using the G *power 3.1.9.2 program [29], the minimum sample size required for multiple linear regression with seven predicting variables, a significance level of 0.05, power of 0.80, and moderate effect size of 0.15 was calculated to be 107. In anticipation of an approximately 80% retrieval rate, a total of 135 nurses were asked to participate in the survey, and responses were obtained from 120 nurses (88.8%).

This study was approved by the Ethical Committee at Eulji University (Approval No. EU 18-1). Questionnaires including a statement about this survey were distributed through the department of nursing at each institution according to the institutions’ policy. Researchers visited each ward during three shifts and explained the purpose of the study to the nurses. Nurses who wanted to participate voluntarily provided signed, informed consent to participate and completed the self-administered questionnaire, which was collected in a sealed envelope anonymously. From those who did not want to participate, the blank questionnaires sealed in the envelopes were collected and submitted anonymously. Data collection was conducted from January to February 2018.

### 2.2. Measures

#### 2.2.1. Chronotype and SJL

In the present study, chronotypes were determined based on the scores of the Morningness–Eveningness Questionnaire-K (MEQ-K) [30,31]; a higher score indicated a morning type and a lower score indicated an evening type. Individuals can be classified into five specific types: Definite evening (16–30), moderate evening (31–41), neither (42–58), moderate morning (59–69), and definite morning (70–86) [31]. The reliability of the MEQ, as assessed using Cronbach’s α, was 0.82 at the time of development. Cronbach’s ⍺ of the MEQ-K is 0.77 [31] and it was 0.72 in this study.

In this study, SJL was the absolute difference between mid-sleep-free days and mid-sleep weekdays on the Munich Chronotype Questionnaire (MCTQ): SJL = mid-sleep-free days − mid-sleep weekdays [26] (see also Roenneberg et al. 2012). Workdays refer to night shifts, and days off refer to days off following night shifts. SJL is described in hours and minutes.

#### 2.2.2. Shift Duty Preference and Fatigue

Preference for shift duty was measured using a visual analog scale (VAS) ranging from 1 (no preference) to 10 (high preference). Fatigue during day shift, evening shift, and night shift were measured using a VAS, ranging from 1 (no fatigue) to 10 (severe fatigue).

#### 2.2.3. Sleep Quality

In this study, overall sleep quality refers to the average sleep quality over a period of one month and was measured using the Pittsburgh Sleep Quality Index-Korean (PSQI-K) [32]. The PSQI-K is a self-reported questionnaire about the quality of sleep and degree of sleep disorder in the preceding month. It consists of 19 items and 7 sleep components: 1 item for subjective sleep quality, 2 items for sleep latency, 2 items for sleep duration, 2 items for usual sleep efficiency, 9 items for sleep disturbance, 1 item for use of sleeping pills, and 2 items for daytime dysfunction. Each component was scored on a scale of 0–3. The PSQI score was calculated according to the formula provided in the instructions of the Sleep Medicine Institute, University of Pittsburgh, USA. A summary score for overall sleep quality can be calculated by adding these components together, yielding scores ranging from 0 to 21. PSQI summary scores that are greater than 5 are generally used to indicate poor sleep, and we adopted this same threshold in our study [33]. The reliability of the PSQI and PSQI-K as measured using Cronbach’s ⍺ were 0.83 and 0.84, respectively, at the time of development, and was 0.77 in this study.

Sleep quality during night shifts was measured using the Verran and Snyder-Halpern (VSH) Sleep Scale [34], which consists of nine items. Eight of these items concerned sleep duration, sleep depth, sleep latency, mood when waking up from sleep, method of waking from sleep, and satisfaction with sleep, and each item was measured using a 10 cm VAS. Each centimeter was converted to a score of 1, and the total score ranged from 0 to 80. A higher score indicates better sleep quality. The final item had a score of 1 for every 1 cm. The final item was an open-ended question asking the participant to describe the reason they could not sleep well. The reliability of the tool as measured using Cronbach’s α was 0.82 and it was 0.76 in this study.

### 2.3. Data Analysis

The data were analyzed using SPSS 24 (IBM, New York, NY, USA). Descriptive statistics were used to analyze the participants’ general characteristics and sleep components. To determine the associations between SJL, chronotype, and sleep components, Pearson’s correlation analyses were conducted. Finally, a multiple regression analysis was conducted to identify the impact of SJL, chronotype, and general characteristics on sleep quality. The Kolmogorov–Smirnov test was used to verify the normality, and the Breusch–Pagan test was used to verify the homoscedasticity of residuals for goodness-of-fit in the regression model.

## 3. Results

### 3.1. Demographic and Work-Related Characteristics of Participants

The mean age of the participants was 25.23 years, and the majority of the participants were women. The total clinical career was 45.32 years, and most participants worked in a general ward. Fifty-four participants claimed that night shifts made sleeping more difficult, while 53 stated that day shifts made sleeping more difficult. Fifty-one did not prefer night shifts, and 76 stated that night shifts are “hard.” The participants’ demographic and work-related characteristics are summarized in Table 1.

### 3.2. Chronotype and SJL

#### 3.2.1. Chronotypes

The chronotype score ranged from 28 to 68 in this study, with a mean score of 40.83. The majority of the participants were moderate evening types, followed by neither type, definite evening types, and moderate morning types. No participants were definite morning types (Table 2).

#### 3.2.2. Sleep Circadian Parameters Including SJL

The mean SJL during night shifts was 2 h and 3 min, and the mean daily sunlight exposure during night shifts was 1 h 44 min. The mean sleep duration during night shifts was 6 h 10 min (Table 2).

### 3.3. Sleep Quality

The mean overall sleep quality score was 8.31. Based on the PSQI-K scores, 104 participants had poor overall sleep quality. The mean sleep duration over a month was 6 h 53 min among the participants, and the mean sleep quality during night shifts, as assessed using the VSH Sleep Scale, was 38.40 (Table 2).

### 3.4. Correlations among Variables

Table 3 shows the correlations among participants’ work-related characteristics, chronotypes, SJL, and sleep quality. Morning chronotypes highly preferred day shifts and had low fatigue during day shifts. SJL was positively correlated with overall sleep quality, suggesting that overall sleep quality decreases with increasing SJL. Further, overall sleep quality decreased with increasing day-shift fatigue. The overall sleep quality was better among those who preferred night shifts and those with increasing sleep quality during night shifts.

A multiple regression analysis using the enter method was performed to identify the predictors of sleep quality. The standardized residuals satisfied the assumption of normality on the Kolmogorov–Smirnov test and the assumption of equal variance on the Breusch–Pagan test, thereby confirming a good fit of the regression model (Table 4); for the regression model, we selected the predicting variables that were correlated to overall sleep quality (Table 3) and that were indicated to influence sleep quality in previous studies [20,35]. This analysis used chronotypes, SJL, VSH scores, age, and day-shift fatigue as the independent variables and PSQI scores as the dependent variable. The SJL, sleep quality during night shifts, and day-shift fatigue were found to significantly influence the overall sleep quality.

The most potent predictor of overall sleep quality was sleep quality during night shifts (β = −0.41, *p* = <0.001), followed by day-shift fatigue (β = 0.24, *p* = 0.006) and SJL (β = 0.16, *p* = 0.045); they explained 24.7% of the variance.

## 4. Discussion

In this study, we measured SJL arising from night shifts among nurses. That is, we measured the SJL occurring as a result of a special work schedule, in which nurses work throughout the night, return home in the morning, sleep, and wake up again in the afternoon for another night shift. The SJL was an average of approximately 2 h 3 min, which is greater than the 1 h and 36 min and 1 h and 25 min measured in previous studies on the general population [21,25], but similar to the 2 h 6 min found in a study that investigated jetlag among night-shift workers in the same situation [36]. While the external clock of social time is the same for everyone, internal time differs substantially across individuals, as shown here [26].

Evening and neither chronotypes accounted for 99% of the participants, and chronotype was not correlated with SJL experienced during night shifts. Furthermore, while chronotypes were not correlated with overall sleep quality and sleep quality during night shifts, SJL had a mild negative correlation with overall sleep quality, suggesting that overall sleep quality deteriorated with increasing SJL, which is consistent with past findings [21]. In a study on rotating shift nurses [17], morning chronotypes had greater SJL and poorer sleep quality while working night shifts. Further, a study of 388 nurses working day and night shifts [37] reported that nurses closer to a morning chronotype show better sleep adjustment to day shifts but low sleep adjustment to night shifts, which is likely due to the greater SJL experienced by morning chronotypes during night shifts. Recently, a study on patients with a sleep disorder and the general population reported that SJL mediates the impact of chronotype on the difference in sleep quality between weekdays and weekends [20], and similar studies are needed on rotating shift nurses.

In a study on the relationship between chronotypes and sleep quality in female rotating shift nurses [17], evening types had 6.6 times greater odds of having a sleep disorder compared to morning types. In general, the morningness phenotype is known to increase with advancing age, following a peak in the eveningness phenotype in the late teens to early 20s [26]. In our study, the mean age of the participants was 26.23, and only one participant was a moderate morning type. Thus, it was not easy to differentiate sleep quality according to chronotype. The participants had a similar level of SJL because most cases were of either type or moderately evening type, without many extreme cases. Regarding chronotypes by age, morning types were twice as frequent in older age groups, with the older age group having only one-third as many evening types compared to the younger group. In addition, sleep quality was higher in the older morning types [17]. We speculate that morning types were nearly absent because our participants worked in a general hospital in a large city and, thus, primarily consisted of younger nurses, which may have contributed to the small difference in sleep quality according to chronotype. However, SJL was correlated with sleep quality, presumably due to the fact that SJL, as opposed to inherent chronotype differences, is a more sensitive indicator of the discrepancy between individuals’ biological and social clocks [26].

We measured overall sleep quality over a month and sleep quality during night shifts. The mean overall monthly sleep quality as measured using the PSQI-K was 8.31, which markedly exceeded the cutoff for sleep disorder. Thus, participants of this study were confirmed to have poor sleep quality, which is consistent with the results of a previous study that compared sleep quality between rotating shift nurses and non-rotating shift nurses [38]; this previous study demonstrated that shift workers had a markedly poorer sleep quality compared to non-shift workers, which supports previous findings that shift work disturbs the circadian rhythm and, thus, deteriorates sleep quality [39]. Furthermore, sleep quality is poor regardless of nurses’ shift schedules [40], highlighting the need to identify and intervene in the factors that affect sleep quality among rotating shift nurses. The sleep quality of nurses working in shift affects the working performance, and the degree of disturbance was significantly higher when sleep quality during the night shift is poor [35]. In the preceding study, nurses complained of difficulty in sleeping during night shifts in particular [14], even though the number of night shifts was the least compared to other shifts, to understand the factors affecting the overall sleep quality of shift workers, we should find the influence of sleep quality during night shift. The mean sleep quality during night shifts, measured using the VSH Sleep Scale, was 38.40, which is similar to that reported by a previous study on nurses who work 2–3 consecutive night shifts with an average of 6–7 night shifts a month [41], and these results show that subjective sleep quality is reduced during night shifts.

In this study, age, chronotypes, day-shift fatigue, SJL, and sleep quality during night shifts were the predictors of the overall monthly sleep quality among rotating shift nurses, and this model explained 24.7% of the variance in sleep quality. These findings are supported by recent review studies on the biological mechanisms of circadian rhythm disturbances caused by shift work. According to a previous review article [42] shift work may negatively affect a series of gene expressions and factors majorly involved in circadian rhythm and sleep homeostasis, therefore, shift work disrupts normal sleep. In a previous study that applied a chronotype-adjusted shift schedule to 8-h rotational shift workers in consideration of SJL (remove night shifts for definitely morning types, remove morning shifts for definitely evening types) for five months, SJL decreased by 1 h overall, and the self-reported sleep duration and quality significantly increased [43]. In other words, taking chronotypes into consideration when planning shift schedules for shift workers such that SJL can be reduced, or using a shift schedule that leads to the least significant SJL, has a positive impact on sleep quality. In recent years, there have been active attempts in the field of medicine to reduce SJL, as opposed to passively accepting it as inevitably caused by chronotypes [44]. In light of such attempts, shift schedules that can lower SJL, which have been confirmed to influence sleep quality, must be implemented for rotating shift nurses.

In this study, we investigated the relationship between chronotypes, SJL, and sleep quality during night shifts to identify the predictors of sleep quality in rotating shift nurses. However, the study does have limitations. For example, the small number of subjects and the lack of extreme chronotypes among the studied subjects could be related to the lack of evidence of a correlation between sleep quality and chronotype (MEQ score). In addition, the vast majority of the participants were women in their 20s and 30s, which limits the generalizability of the findings. We did not use quota or cluster sampling in recruiting participants for the study, therefore, selection bias cannot be excluded as a limitation of the study. In particular, in the process of including only subjects who were interested in this study and wanted to participate, most subjects were limited to those in their 20s and 30s, thus narrowing the distribution of chronotypes. Therefore, with an under-representation of morning and extreme morning chronotype, possible social jetlag between different chronotypes was not sufficiently revealed. Furthermore, as this was a cross-sectional study, we could not determine causality. In the future, it would be beneficial to conduct prospective cohort studies. Moreover, we suggest sleep log diary studies to gain deeper insights on sleep quality reported day by day and for studies to use smart devices which can measure and record sleep parameters including sleep quality to have more objective data. Additional studies that address these limitations are needed.

## 5. Conclusions

This cross-sectional study aimed to examine the chronotypes and degree of SJL experienced by nurses working night shifts and identify the predictors of sleep quality in rotating shift nurses, ultimately providing foundational data for developing interventions to improve sleep quality and devise efficient shift schedules for rotating shift nurses. This study indicated that average monthly sleep quality increases with decreasing day-shift fatigue, decreasing SJL, and increasing sleep quality following night shifts. To improve the overall sleep quality for these rotating shift nurses, subsequent studies should explore measures to improve sleep quality during night shifts and develop shift schedules that reduce SJL.

## Figures and Tables

**Table 1 ijerph-18-00047-t001:** Demographic and work-related characteristics of participants (*n* = 120).

Variables	Categories	*n* (%)	M ± SD
Gender	Male	2 (1.7)	
	Female	118 (98.3)	
Age (years)	<30	105 (87.5)	26.23 ± 3.39
	≥30	15 (12.5)	
Health Condition	Good	39 (32.5)	
	Moderate	65 (54.2)	
	Bad	16 (13.3)	
Housemates	Alone (zero)	58 (48.3)	
	Together (1 or more)	62 (51.7)	
Work Unit	Ward	99 (82.5)	
	Intensive care unit	21 (17.5)	
Current unit Clinical Career (months)			37.08 ± 26.87
Total Clinical Career (months)			45.32 ± 32.07
Number of Night Shifts/Month	3–4 times 5–6 times	15 (12.4) 60(50.0)	
	Over 7 times	45 (37.6)	
Shift that most affects sleep *	Day duty *	53 (44.2)	
	Evening duty	13 (10.8)	
	Night duty	54 (45.0)	
Preference for Night Duty	Prefer	21 (17.5)	
	No preference	48 (40.0)	
	Not prefer	51 (42.5)	
Difficulty of Night Duty	Hard	76 (63.3)	
	Moderate	39 (32.5)	
	Not hard	5 (4.2)	

* Nurses chose ‘day shift’ when they could not sleep because they were working the day shift the next day. M: Mean; SD: Standard deviation.

**Table 2 ijerph-18-00047-t002:** Morningness–Eveningness Questionnaire (MEQ) score, Chronotype 5 categories, Circadian rhythm parameters, and quality of sleep of subjects (*n* = 120).

Variable	Category (Reference)	*n* (%)	M ± SD	Min	Max
MEQ (Chronotype)	Extreme evening type (16–30)	7 (5.8)	28.71 ± 0.76	28	30
	Moderate evening type (31–41)	63 (52.5)	36.89 ± 2.80	31	41
	Neither type (42–58)	49 (40.8)	47.08 ± 5.00	42	58
	Moderate morning type (59–69)	1 (0.8)	68.00 ± 0.00	68	68
	Extreme morning type (70–86)	0 (0.0)	-	-	-
Social Jetlag (absolute value)			2:03 ± 2:33	0:00	11:20
Circadian Rhythm Parameters (hours:minutes)	Average light exposure/day		1:44 ± 1:26	0:00	9:12
	Average sleep duration (night shift)		6:10 ± 1:41	2:30	10:11
PSQI (Overall Sleep Quality)	Good quality of sleep (PSQI < 5)	16 (13.3)			
	Poor quality of sleep (PSQI ≥ 5)	104 (86.7)			
	Sleep Duration (hours:minutes)		6:53± 1:43	3:00	14:00
	Total score		8.31± 3.04	3	18
VSH (Sleep Quality during Night Shift)	Total score		38.40 ± 11.46	7	69

VSH: Verran and Snyder-Halpern (VSH) Sleep Scale; MEQ: Morningness–Eveningness questionnaire; PSQI: Pittsburgh Sleep Quality Index; M: mean; SD: Standard deviation.

**Table 3 ijerph-18-00047-t003:** Correlation between work-related characteristics, chronotype, social jetlag (SJL), and sleep quality (*n* = 120).

		Age	Total Clinical Experience	Preference			Fatigue			MEQ	SJL	VSH	PSQI
				Day Shift	Evening Shift	Night Shift	Day Shift	Evening Shift	Night Shift				
		r (*p*)	r (*p*)	r (*p*)	r (*p*)	r (*p*)	r (*p*)	r (*p*)	r (*p*)	r (*p*)	r (*p*)	r (*p*)	r (*p*)
**Age**		1											
Total Clinical Experience		0.869 (<0.001) ***	1										
Preference	Day shift	0.032 (0.725)	−0.092 (0.316)	1									
	Evening shift	0.155 (0.092)	0.118 (0.197)	−0.126 (0.169)	1								
	Night shift	−0.121 (0.187)	−0.182 (0.047) *	−0.033 (0.723)	0.199 (0.029) *	1							
Fatigue	Day shift	0.196 (0.032) *	0.227 (0.013) *	−0.328 (<0.001) ***	0.282 (0.002) **	0.177 (0.053)	1						
	Evening shift	−0.065 (0.480)	−0.134 (0.146)	0.204(0.025) *	−0.285 (0.002) **	0.008 (0.934)	0.112 (0.224)	1					
	Night shift	0.141 (0.124)	0.144 (0.116)	0.134 (0.145)	−0.105 (0.255)	−0.528 (<0.001) ***	0.075 (0.413)	0.139 (0.130)	1				
MEQ		0.098 (0.285)	0.109 (0.235)	0.229 (0.012) *	−0.177 (0.054)	−0.147 (0.110)	−0.310 (0.001) **	−0.029 (0.754)	−0.036 (0.700)	1			
SJL		−0.049 (0.593)	−0.053 (0.566)	−0.042 (0.649)	0.032 (0.726)	0.149 (0.104)	−0.032 (0.726)	−0.042 (0.649)	−0.171 (0.061)	−0.007 (0.936)	1		
VSH		−0.150 (0.103)	−0.171 (0.062)	0.095 (0.300)	0.095 (0.301)	0.303 (0.001) **	−0.159 (0.083)	−0.057 (0.538)	−0.108 (0.241)	−0.016 (0.859)	−0.064 (0.488)	1	
PSQI		−0.046 (0.616)	−0.006 (0.947)	−0.172 (0.060)	−0.097 (0.290)	−0.033 (0.724)	0.245 (0.007) **	0.019 (0.213)	−0.039 (0.672)	0.002 (0.982)	0.187 (0.040) *	−0.439 (<0.001) ***	1

* *p* < 0.05, ** *p* < 0.01, *** *p* < 0.001; VSH: Verran and Snyder-Halpern (VSH) Sleep Scale; MEQ: Morningness–Eveningness questionnaire; PSQI: Pittsburgh Sleep Quality Index.3.5. Factors influencing sleep quality.

**Table 4 ijerph-18-00047-t004:** Influence of monthly average sleep quality.

Independent Variable	B	SE	β	t	*p*	VIF	95% CI	
							Lower	Upper
(constant)	11.15	2.843		3.92	<0.001		5.51	16.78
Age	−0.14	0.074	−0.16	−1.88	0.063	1.09	−0.29	0.01
MEQ score	0.04	0.04	0.09	1.02	0.308	1.14	−0.03	0.11
VSH	−0.11	0.02	−0.41	−5.06	<0.001	1.05	−0.15	−0.07
Social jetlag	0.01	0.00	0.16	2.02	0.045	1.01	0.000	0.01
Day-shift fatigue	0.45	0.16	0.24	2.78	0.006	1.20	0.13	0.77
Adj R^2^ = 0.247								
F = 8.79 ***								

Durbin–Watson = 2.22; Durbin–Watson’s du (upper critical limit) = 1.77, 4-du (lower critical limit) = 2.23; Breusch–Pagan’s χ^2^ = 12.71(*p* = 0.122); Kolmogorov–Smirnov test (Z = 0.61, *p* = 0.856); VSH: Verran and Snyder-Halpern (VSH) Sleep Scale; MEQ: Morningness–Eveningness questionnaire. Dependent variable: PSQI. *** *p* < 0.001.

## Data Availability

The data presented in this study are available on request from the corresponding author and with permission of the Institutional Review Board of Eulji University.

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
