# Peer review of "The Impact of Social Jetlag on Sleep Quality among Nurses: A Cross-Sectional Survey"

_ijerph, 2020, doi:10.3390/ijerph18010047_

Round 1
Reviewer 1 Report
General comments
The MS sounds well written, an interesting study is presented and, in principle, the results are of some relevance.
Nevertheless, in my opinion the study has at least a weakness: the recruitment of participants.
How was the enrollment organized?
“Questionnaires were distributed through the department of nursing at each institution.”
How? How the personnel received information on the research?
Furthermore, a sample of 120 nurses voluntary participated: how are the Authors confident that the a selection bias was avoided?
And, in effect, the possibility of a selection bias is suggested by the age of the participants, relatively narrow and including almost exclusively young nurses (I suppose that the distribution of age of in the nurses working in the two involved general hospital was different, also including senior nurses), as well as by the underrepresentation of morning type chronotype, both, moderate and extreme. This is a pity, as differences in social jetlag between different chronotypes are possible/likely, and a comparison between morning and evening type should be of interests.
The Another ‘d consider and adequately discuss these points in the MS and in the conclusions.
Specific comments
Introduction:
Lines 26-28:
even if epidemiological data supporting a relation between the reported adverse health effects exist, performed studies showed mixed results, and a causal relationship cannot considered adequately proved.
Line 92:
“.. The present study was approved by the institutional review board at E University (Approval 93 No. EU 18-1). :
Is it an Ethics committee? , and what is E University?
Line 184,
“Age and day-shift fatigue were found to significantly differ according to sleep quality.”:
There is something not clear to me in this statement: in Table 4, considering a significance level of 0.05, apparently SJL is significantly correlated (t= 2.02; p= 0.045), while age (t= -1.88; p = 0.06) is not. Can Authors explain to me?
Table 1:
is difficult to follow: I suggest to revise formatting; a revision of spacing can be useful
Furthermore, the Table is very detailed, but are all the provided characteristics needed (especially considering that several are not used in following evaluations/discussion)?
Table 2:
At least for some of the parameters (e.g. the SJL) the SD is very large compared to the mean (I suppose that M is the arithmetic mean, is it? Apparently it’s not declared in the table or caption); was the distribution of data normal?
Table 3
In the version I see, the correct formatting was lost, so reading data is difficult.
Author Response
Response to Reviewer 1
- How was the enrollment organized?
“Questionnaires were distributed through the department of nursing at each institution.”
How? How the personnel received information on the research?
Response: We would like to thank you for your constructive suggestions and feedback. They have greatly improved the overall quality of our manuscript.
Accordingly, we have revised the manuscript to state:
Page 2–3, lines 94–100: Questionnaires including a statement about this survey were distributed through the department of nursing at each institution according to the institutions’ policy. Researchers visited each ward during three shifts and explained the purpose of the study to the nurses. Nurses who wanted to participate voluntarily provided signed, informed consent to participate and completed the self-administered questionnaire, which was collected in a sealed envelope anonymously. From those who did not want to participate, the blank questionnaires sealed in the envelopes were collected and submitted anonymously.
- Furthermore, a sample of 120 nurses voluntary participated: how are the Authors confident that the a selection bias was avoided?
Response: Thank you for your helpful comments. Accordingly, we have added a statement that the selection bias cannot be excluded as a limitation of the study to state:
Page 10, lines 289–290: We did not use quota or cluster sampling in recruiting participants for the study, therefore, selection bias cannot be excluded as a limitation of the study.
- And, in effect, the possibility of a selection bias is suggested by the age of the participants, relatively narrow and including almost exclusively young nurses (I suppose that the distribution of age of in the nurses working in the two involved general hospital was different, also including senior nurses), as well as by the underrepresentation of morning type chronotype, both, moderate and extreme. This is a pity, as differences in social jetlag between different chronotypes are possible/likely, and a comparison between morning and evening type should be of interests.
The Another ‘d consider and adequately discuss these points in the MS and in the conclusions.
Response: As you recommended, we have made additions in manuscript, including the conclusion section:
Page 10, lines 290–292:
In particular, in the process of including only subjects who were interested in this study and wanted to participate, most subjects were limited to those in their 20s and 30s, thus narrowing the distribution of chronotypes.
Page 10, line 302–305:
However, since it was an under-representation of morning and extreme morning chronotype, possible social jetlag between different chronotypes was not sufficiently revealed, which is thought to be due to selection bias.
- Lines 26-28: even if epidemiological data supporting a relation between the reported adverse health effects exist, performed studies showed mixed results, and a causal relationship cannot considered adequately proved.
Response: Thank you very much. As you recommended, we revised the part in the manuscript:
Page 1, lines 26–29:
Shift work disturbs workers' circadian rhythms. Therefore, it may reduce effective performance [1] and increase the incidence of occupational accidents and injuries, coronary heart disease, strokes, type 2 diabetes, cancer, obesity, depression, and all-cause mortality [2,3].
- Line 92: “.. The present study was approved by the institutional review board at E University (Approval No. EU 18-1).
Is it an Ethics committee? and what is E University?
Response: Yes. It is an Ethics committee. We had misunderstood that we have to blind university name as stating in initial. We have revised it as given below:
Page 2, line 93–94:
This study was approved by the Ethical Committee at Eulji University (Approval No. EU 18-1).
- Line 184, “Age and day-shift fatigue were found to significantly differ according to sleep quality.”: There is something not clear to me in this statement: in Table 4, considering a significance level of 0.05, apparently SJL is significantly correlated (t= 2.02; p= 0.045), while age (t= -1.88; p = 0.06) is not. Can Authors explain to me?
Response: We apologize for our oversight. It is not age, but the SJL. We revised as below:
Page 8, lines 193–194:
The SJL, sleep quality during night shifts, and day-shift fatigue were found to significantly influence the overall sleep quality.
- Table 1: is difficult to follow: I suggest to revise formatting; a revision of spacing can be useful
Furthermore, the Table is very detailed, but are all the provided characteristics needed (especially considering that several are not used in following evaluations/discussion)?
Response: We appreciate this suggestion.
Page 4:
We revised Table 1 spacing. As you pointed out, all the provided characteristics were not needed, therefore, we removed education and religion and rearranged the table for readability.
- Table 2: At least for some of the parameters (e.g. the SJL) the SD is very large compared to the mean (I suppose that M is the arithmetic mean, is it? Apparently it’s not declared in the table or caption); was the distribution of data normal?
Response: Yes, right. It is arithmetic mean.
Page 5, line 167: Accordingly, we have revised the manuscript. M: mean; SD: Standard deviation
The distribution of SJL was not a normal distribution. It has given the numbers for readers to help understand when comparing the previous studies’ results, but if you suggest to replace it with the median, we will revise again accordingly.
- Table 3: In the version I see, the correct formatting was lost, so reading data is difficult.
Response: Page 7: Accordingly, we have formatted the Table 3 correctly.
Reviewer 2 Report
Review of the manuscript “The impact of social jetlag on sleep quality among nurses: A cross-sectional survey”
This study focuses on the relationship between chronotype, social jetlag and sleep quality. In a correlational study among nurses the Authors obtained interesting results. The introduction is generally well structured, the measures and analysis are adequately described. I have some minor concerns and additional suggestions for future studies.
Firstly, the sample is specific (very young, rather singles, with no children), thus I would ask the Authors to add information if the sample is comparative to the general nurses sample in Korea (if generally in Korea nurses are rather young women). It would be especially interesting and important to conduct such a study among people +40. We may expect that the results would show stronger effects.
In the discussion on future research the Authors may mention not only longitudinal but also diary studies to have deeper insight on sleep quality reported day by day. Moreover, apart from questionnaire measurements of sleep quality the Authors may consider using smart watches with sleep quality analysis measurements to have more objective data.
After introducing these improvements it will be possible to publish the manuscript in International Journal of Environmental Research and Public Health.
Author Response
Response to Reviewer 2
- Firstly, the sample is specific (very young, rather singles, with no children), thus I would ask the Authors to add information if the sample is comparative to the general nurses sample in Korea (if generally in Korea nurses are rather young women). It would be especially interesting and important to conduct such a study among people +40. We may expect that the results would show stronger effects.
Response: We would like to thank you for your constructive suggestions and feedback. They have greatly improved the overall quality of our manuscript.
Considering ethical issues, as a result of voluntary willingness to participate in research participation, unfortunately, the age group tends to focus on the younger participants. This is a limitation of this study and is described in the discussion and conclusions. In addition, in order to compensate for this, it is proposed to perform quota sampling considering the age group when collecting data in future studies.
Page 10, lines 289–292 in the discussion section: Accordingly, we have revised manuscript as below:
We did not use quota or cluster sampling in recruiting participants for the study, therefore, selection bias cannot be excluded as a limitation of the study. In particular, in the process of including only subjects who were interested in this study and wanted to participate, most subjects were limited to those in their 20s and 30s, thus narrowing the distribution of chronotypes.
Page 10, lines 302–305 in the conclusion section:
However, since it was an under-representation of morning and extreme morning chronotype, possible social jetlag between different chronotypes was not sufficiently revealed, which is thought to be due to selection bias.
- In the discussion on future research the Authors may mention not only longitudinal but also diary studies to have deeper insight on sleep quality reported day by day. Moreover, apart from questionnaire measurements of sleep quality the Authors may consider using smart watches with sleep quality analysis measurements to have more objective data.
After introducing these improvements it will be possible to publish the manuscript in International Journal of Environmental Research and Public Health.
Response: Thank you very much for your insightful comments.
Page 10, lines 294–296: Accordingly, we have revised this part:
Moreover, we suggest sleep log diary studies to gain deeper insights on sleep quality reported day by day and for studies to use smart devices which can measure and record sleep parameters including sleep quality to have more objective data.
Reviewer 3 Report
Authors investigated for the relationship between social jetlag and sleep quality among nurses. Results showed that low VSH score, longer social jetlag time, higher day-shift fatigue were significantly associated with higher PSQI score.
Although there are some interesting points, the manuscript should be improved to be published.
- Please clarify the term for the concept and the tool which was used to measure the concept in the whole manuscript. For example, "PSQI score" and "sleep quality", "MEQ score" and "chronotypes", and "VSH score" and "sleep quality after night shifts." In the regression model (line 184), dependent variable was PSQI score, not sleep quality (sleep quality itself does not have specific value).
- Please describe how to select the predicting variables for the regression model.
- It seems to be a kind of circular reasoning to say the sleep quality during night shifts predicts sleep quality. Please justify this in the discussion section, or remodel the regression model.
- The meaning is not clear: (line 184-185) "Age and day-shift fatigue were found to significantly differ according to sleep quality."
- (line 190-199) It does not need to be described in the manuscript.
- Please discuss about the biological underlying mechanisms that support the results of the study in the discussion section.
- (line 264) Although authors investigated for some more variables, the main outcome of this study was the relationship between chronotypes and sleep quality. Social jetlag and sleep quality during night shifts in this study were specific aspects from chronotypes and shift work. It is hard to say that this was the first study about the relationship between chronotypes and sleep quality. This sentence should be deleted.
- (line 275 and abstract) The relationship between sleep quality and other variables cannot be confirmed in this cross-sectional study.
- Why did authors collect data on respondents' religion?
Author Response
Response to Reviewer 3
- Please clarify the term for the concept and the tool which was used to measure the concept in the whole manuscript. For example, "PSQI score" and "sleep quality", "MEQ score" and "chronotypes", and "VSH score" and "sleep quality after night shifts." In the regression model (line 184), dependent variable was PSQI score, not sleep quality (sleep quality itself does not have specific value).
Response: We would like to thank you for your constructive suggestions and feedback. They have greatly improved the overall quality of our manuscript.
Accordingly, we have clarified the term for the concept and the tool which was used to measure the concept in the whole manuscript, especially Page 8, lines 191–193:
This analysis used chronotypes, SJL, VSH scores, age, and day-shift fatigue as the independent variables and PSQI scores as the dependent variable.
- Please describe how to select the predicting variables for the regression model.
Response: Page 8, lines 188–191: Accordingly, we have revised:
A multiple regression analysis using the enter method was performed to identify the predictors of sleep quality (Table 4); for the regression model, we selected the predicting variables which were correlated to overall sleep quality (Table 3) and which were indicated to influence sleep quality in previous studies [20,35].
- It seems to be a kind of circular reasoning to say the sleep quality during night shifts predicts sleep quality. Please justify this in the discussion section, or remodel the regression model.
Response: Thank you very much for your comments and request for justification. Page 9, lines 258–263: Accordingly, we have revised the text as follows:
The sleep quality of nurses working in shift affects the working performance, and the degree of disturbance was significantly higher when sleep quality during the night shift is poor [35]. In the preceding study, nurses complained of difficulty in sleeping during night shifts in particular [14], even though the number of night shifts was the least compared to other shifts, to understand the factors affecting the overall sleep quality of shift workers, we should find the influence of sleep quality during night shift.
- The meaning is not clear: (line 184-185) "Age and day-shift fatigue were found to significantly differ according to sleep quality."
Response: Thank you very much for your comments and suggestion to improve clarity. We apologize for our oversight. It is not age but SJL. We revised as below:
Page 8, lines 193–194:
The SJL, sleep quality during night shifts, and day-shift fatigue were found to significantly influence the overall sleep quality.
5.(line 190-199) It does not need to be described in the manuscript.
Response: Accordingly, we have deleted this part. Page 8, lines 200-209
- Please discuss about the biological underlying mechanisms that support the results of the study in the discussion section.
Response: Thank you very much for your suggestion. As you recommended, we have revised the section as below:
Page 9, lines 270–274:
These findings are supported by recent review studies on the biological mechanisms of circadian rhythm disturbances caused by shift work. According to previous review article [42] shift work may negatively affect a series of gene expressions and factors majorly involved in circadian rhythm and sleep homeostasis, therefore, shift work disrupts normal sleep.
7.(line 264) Although authors investigated for some more variables, the main outcome of this study was the relationship between chronotypes and sleep quality. Social jetlag and sleep quality during night shifts in this study were specific aspects from chronotypes and shift work. It is hard to say that this was the first study about the relationship between chronotypes and sleep quality. This sentence should be deleted.
Response: Yes. You are right. Thank you again. Accordingly, we have revised the statement as below:
Page 10, line 284–286:
In this study, we investigated the relationship between chronotypes, SJL, and sleep quality during night shifts to identify the predictors of sleep quality in rotating shift nurses.
- (line 275 and abstract) The relationship between sleep quality and other variables cannot be confirmed in this cross-sectional study.
Response: Thank you for your helpful comments. Accordingly, we have revised it in the manuscript and abstract as below:
Abstract (Page 1, lines 21–22): The study concluded that overall sleep quality may increase with decreasing day-shift fatigue, decreasing SJL, and increasing sleep quality on night shifts.
Page 10, lines 305–306: This study indicated that average monthly sleep quality increases with decreasing day-shift fatigue, decreasing SJL, and increasing sleep quality following night shifts.
- Why did authors collect data on respondents' religion?
Response: We appreciate this question. We collected this information because we thought that religion could affect their sleep when nurses choose to attend early morning services such as Holy Mass. However, contrary to expectations, there was no difference in the overall sleep quality according because of religion (F=0.234, p=.873). In line with the opinion of Reviewer 1, we have deleted those parts from Table 1 to make it simpler.
As you and reviewer 1 pointed out that all the provided characteristics were not needed, we have removed education and religion and rearranged Table 1 for better readability.
Round 2
Reviewer 3 Report
Thank you for considering my comments.
In my opinion, the manuscript has been improved substantially.
I appreciate your valuable work.
Author Response
Thank you for your kind words.